# On ATG4B as Drug Target for Treatment of Solid Tumours—The Knowns and the Unknowns

**DOI:** 10.3390/cells9010053

**Published:** 2019-12-24

**Authors:** Alexander Agrotis, Robin Ketteler

**Affiliations:** MRC Laboratory for Molecular Cell Biology, University College London, London WC1E 6BT, UK; alex.agrotis@hotmail.co.uk

**Keywords:** autophagy, ATG4, pancreatic ductal adenocarcinoma (PDAC), drug screening, small molecule compound, screening assay, biomarker

## Abstract

Autophagy is an evolutionary conserved stress survival pathway that has been shown to play an important role in the initiation, progression, and metastasis of multiple cancers; however, little progress has been made to date in translation of basic research to clinical application. This is partially due to an incomplete understanding of the role of autophagy in the different stages of cancer, and also to an incomplete assessment of potential drug targets in the autophagy pathway. While drug discovery efforts are on-going to target enzymes involved in the initiation phase of the autophagosome, e.g., unc51-like autophagy activating kinase (ULK)1/2, vacuolar protein sorting 34 (Vps34), and autophagy-related (ATG)7, we propose that the cysteine protease ATG4B is a bona fide drug target for the development of anti-cancer treatments. In this review, we highlight some of the recent advances in our understanding of the role of ATG4B in autophagy and its relevance to cancer, and perform a critical evaluation of ATG4B as a druggable cancer target.

## 1. Introduction

Autophagy is a cellular stress response that has been identified as a target pathway for intervention in various diseases such as neuro-degenerative disorders, pathogen infection, and various types of cancer [1]. Importantly, autophagy allows cells to adapt to their metabolic needs. This is of particular importance in cancer cells, which constantly encounter stresses such as hypoxia or nutrient limitation, and thus have to evolve to manage these stresses and survive. Autophagy mediates the breakdown of cellular components, including proteins, lipids, sugars, and whole organelles. These structures are taken up by the autophagosome and delivered by the autophagy machinery to the lysosome, where these components are broken down into building blocks, such as amino acids, fatty acids, sugars, and nucleotides. These products can fuel the metabolic demand of cancer cells by feeding into downstream ana- and catabolic reactions. In addition, autophagy limits the effects of damaged organelles, in particular mitochondria as a consequence of reactive oxygen species.

The role of autophagy in cancer is complex and dependent on the stage, type, and status of the tumour [2,3,4]. For instance, it is thought that a growing tumour depends heavily on autophagy for survival to mitigate problems with nutrient limitation, hypoxia, and metabolic limitations. On the other hand, autophagic degradation can remove damage that occurs as a consequence of tumour DNA instability or mitochondria malfunction. For instance, it has been noted that ATG5 defective cells can induce the formation of spontaneous liver tumours [5]. Since these are mostly benign adenomas, it was suggested that progression to a malignant state may be dependent on autophagy [6]. This was supported by studies demonstration that K-Ras-driven tumours are highly dependent on autophagy for survival [7,8]. Overall, these findings suggest that autophagy may be tumour suppressive at the tumour initiation stage, while tumour promoting at later stages.

Cross-cancer profiling of autophagy gene expression has been performed and has identified key autophagy genes that are either altered specifically in cancer, or unaltered in cancer [9,10]. For instance, a high level of autophagy gene expression alteration was found in breast cancer, suggesting that these copy number alterations expose vulnerabilities for anti-cancer treatment. However, on the other hand, it is possible that certain cancers maintain a tight balance on autophagy gene expression to ensure proper functioning and survival/growth. This is certainly the case for glioblastoma, where autophagy gene expression is remarkably unaltered [9].

It is generally thought that inhibition of autophagy provides a benefit as anti-cancer strategy [4,11]. The crucial step for inhibition is to block either the formation of an autophagosome or to block fusion of the autophagosome with the lysosome. Such inhibition can occur at various points in the autophagy pathway (Figure 1).

The formation of an autophagosome generally occurs in four distinct steps: Initiation, elongation, and closure, followed by lysosomal fusion. Initiation is regulated by the ULK1/2 and vps34/phosphatidyl-inositol kinase class III protein kinase complexes that lead to phosphorylation of downstream molecules, resulting in the formation of a cup-shaped phagophore that further elongates and closes prior to fusion with the lysosome.

A key step in the formation of an autophagosome is the processing of LC3/GABARAP (microtubule-associated protein 1 light chain 3/gamma aminobutyric acid receptor-associated protein), an ubiquitin-like protein that links the autophagosome membrane to the autophagy machinery. LC3/GABARAP undergoes three distinct processing steps (Figure 2): (a) Proteolytic cleavage of the C-terminus by an ATG4 protease family member to generate LC3/GABARAP-I; (b) conjugation of LC3/GABARAP-I with phosphatidylethanolamine (PE) by ATG3/7 to generate LC3/GABARAP-II; and (c) hydrolysis of the PE-anchor and recycling of LC3/GABARAP-I by an ATG4 family member.

It is emerging that the activity of ATG4B is tightly controlled by post-translational modifications, including phosphorylation mediated by ATG1/ULK1 [12,13], AKT1 [14], and serine/threonine protein kinase MST4 [15]. For instance, phosphorylation of ATG4B by ULK1 on serine 316 results in a reduction of LC3-II hydrolysis [12], allowing the autophagosome to close before premature deconjugation of LC-II from the autophagosome membrane. Similarly, members of the LC3/GABARAP family can be phosphorylated by Tank-binding kinase 1 (TBK1), which renders them resistant to ATG4-mediated deconjugation [16]. These post-translational mechanisms of regulation suggest a spatio-temporal control of key autophagy genes to enable completion of autophagosome maturation. In addition to its role in autophagy, some non-autophagy roles for the LC3/GABARAP proteins have been described. For instance, LC3/GABARAP can be attached to non-autophagosome, single-membrane compartments and have a role in LC3-associated phagocytosis (LAP) [17,18] and LC3-associated endocytosis (LANDO) [19]. Of note, proteins are also tagged with LC3/GABARAP as a ubiquitin-like post-translational modification (“LC3/GABARAP-ylation”) [20]. LAP, LANDO, and LC3/GABARAP-ylation are, to some degree, regulated by the ATG4 family of proteins, and thus the development of ATG4B inhibitors may impact on these processes in addition to the effects on canonical autophagy. Recently, it has been shown that ATG4B proteolytic activity is also essential for replication of enterovirus 71 and can process viral polyprotein, possibly indicating that other cellular targets for this protease exist [21].

Both academic labs and pharmaceutical companies have proposed and explored the possibility to develop small molecule inhibitors for targeting autophagy in cancer. This has been done using phenotypic screening trying to block the formation or fusion of auto-lysosomes [22,23], which has been very successful in identifying research tool compounds for investigating the pathway in more detail. Yet, with the exception of chloroquine and its derivatives, none of these compounds have translated into drug discovery programmes. Several clinical trials were completed or are on-going using chloroquine or the more stable analogue hydroxychloroquine for treatment of various forms of cancer, often in combination with first line chemotherapeutic agents [4,24,25,26,27]. However, there are limitations in the use of hydroxychloroquine, since it inhibits lysosome function, in addition to its effects on autophagy. Until recently, the mechanism of action of hydroxychloroquine and derivatives has been unclear, but it is now thought to inhibit the lysosomal enzyme palmitoyl-protein thioesterase 1, which may account for the observed effects on autophagy in cancer [28]. However, the potency and pharmacology of hydroxychloroquine, as well as its inherent basicity, are a concern for drug development [4,29,30]. Furthermore, chloroquine can induce LAP [31], thus complicating the assessment of autophagy-specific effects. 

Some inhibitors targeting autophagy have been described. Considerable work has been done to develop inhibitors for ULK1/2, ATG7, and vps34, but most of these have not been further developed for anti-cancer treatments, mainly because of toxicity. Mice deficient in core autophagy components such as ATG5 or ATG7 die shortly after birth, during the neonatal starvation period [32]. Systemic whole-body knockout of ATG7 in adult mice results in many histopathological abnormalities and a reduced lifespan [33]. Typically, these mice die from neurodegeneration or bacterial infection. In addition, liver damage, splenomegaly, testicular degeneration, low white adipose tissue, muscle degeneration, pancreatic damage, and fatal hypoglycemia upon fasting were observed [33]. Neuronal dysfunction resulting in an inability to suckle has been proposed as the key contributor to death in ATG5 knockout animals. Interestingly, neuron-specific expression of ATG5 can rescue these mice [34]. Adult ATG5 knockout mice that express ATG5 in the brain show similar abnormalities as the ATG7 knockout (except the neurodegeneration phenotype), but in addition also display abnormalities related to inflammatory responses, in particular in the kidney, liver, spleen, and lymph nodes, as well as atrophy of sex organs and anaemia, resulting from iron deficiency [34]. Mice deficient in ATG5 and liver-specific deletion of ATG7 develop benign liver adenomas [5], supporting the notion that autophagy may be tumour suppressive at the beginning of tumourigenesis. In a mouse model with doxycycline-inducible shRNA-mediated depletion of ATG5, hepatomegaly and liver damage was observed [35]. These effects were reversible, but resulted in hepatic fibrosis. These studies confirm that autophagy inhibition can promote benign tumour growth, but additional oncogenic insults may be required to result in organ toxicity [4,36].

Here, we propose the autophagy-specific ATG4 proteases as a druggable class of enzymes that can overcome these limitations. The ATG4 proteases mediate processing of ubiquitin-like Atg8/LC3/GABARAP proteins involved in the formation and maturation of autophagosomes, the functional units of the autophagy pathway [37]. Here, we will perform a critical appraisal of ATG4B as drug target in cancers and discuss tools and reagents available for drug development. 

## 2. ATG4B as Drug Target

Mammalian cells show a high degree of functional redundancy in the ATG4/ATG8 system. In mammals, there are four ATG4 isoforms that contribute to processing of the seven LC3/GABARAP family members. Each family member has a different preference for LC3/GABARAP substrates and a difference in the recognition and cleavage of the pro-peptide substrate as opposed to the lipidated LC3/GABARAP-II version has been proposed [38]. The in vitro activity of ATG4 family members has been established using tagged LC3/GABARAP constructs or cellular overexpression and using semi-quantitative methods such as Western blotting for determining the kinetic parameters [39]. ATG4B is considered the major isoform responsible for cleaving LC3/GABARAP proteins. In fact, ATG4B has the capability to cleave all seven members of the LC3/GABARAP family [40,41]. ATG4A, on the one hand, preferentially processes GABARAP proteins in vitro, although with much slower kinetics compared to ATG4B [40]. On the other hand, ATG4C and ATG4D displayed very little activity towards LC3/GABARAP proteins in vitro [40], and it was shown that caspase 3-mediated cleavage of ATG4D enhanced its activity [42]. 

Until recently, the contribution of each ATG4 isoform to processing of one of the seven LC3/GABARAP members in cells was not clear. Studies using catalytic inactive ATG4B C74A have hinted at a dominant role for ATG4B in autophagosome formation [43], based on findings that incomplete autophagosomes accumulate in such cells. However, this model is complicated by the fact that ATG4B C74A very strongly binds to its substrates, acting as a sink that depletes LC3/GABARAPs from the cytoplasm. More recently, cell lines edited by clustered regularly interspaced palindromic repeats (CRISPR) technology, where one or a combination of all four mammalian ATG4 family members have been deleted, were reported [37,38]. In addition, a cell line with deletion of all major LC3/GABARAP proteins has been produced and studied with regards to their resistance to external stresses [44,45]. These studies have largely confirmed in vitro findings, supporting the notion that ATG4B is the dominant isoform involved in LC3/GABARAP processing. 

All four ATG4 isoforms are ubiquitously expressed in mammals, and some tissue-specific differences have been proposed for the ATG4 family members in mice, though not confirmed. Tissue-specific expression of ATG4B was also measured in rats and it was found that ATG4B mRNA was ubiquitous in the rat tissues analysed, with particularly high levels in brain and testicular tissue [46]. The expression levels of ATG4 proteins are regulated at the transcriptional and post-translational level. Various transcription factors, including forkhead box O3 (FOXO3A), early growth response 1 (Egr-1), and CCAAT/enhancer binding protein (C/EBP)beta, can enhance ATG4B expression [47,48,49]. Interestingly, several autophagy genes are direct targets of the tumour suppressor p53, and it has been shown that p53 reduces ATG4C/D levels [50]. In addition, miRNAs exist that down-regulate the expression of various ATG4 members, including miR144-3p [51] and miR24-3p [52] for ATG4A; miR-34A [53], let7i [54], and miR665-3p [55] for ATG4B; miR376a and miR376b for ATG4C [56]; and miR101 for ATG4D [57,58]. ATG4B protein stability is regulated by ubiquitin-mediated degradation [59]. However, overall expression of ATG4 proteins in healthy cells is quite stable and ubiquitous. ATG4B undergoes several types of post-translational modifications under multiple conditions [60], including ubiquitination [59], O-GlcNAcylation [61], S-nitrosylation [62], caspase-mediated proteolysis [42,63], redox mechanisms [64,65,66], and phosphorylation [12,13,14,15,67].

The ATG4 proteins are cysteine proteases with a catalytic domain that is conserved among the papain family of cysteine proteases, and contains a classical catalytic triad of Cys74, Asp278, and His280. Proteases in general have been intensely studied as drug targets, and the identification of selective inhibitors for many of these family members has been described, from the development of angiotensin converting enzyme inhibitors in the early 1980s to human immunodeficiency virus (HIV) protease inhibitors [68]. Typically, chemical inhibition was achieved by the use of nitrile or electrophilic warheads, although irreversibility of such modes of inhibition can be a concern. Typical warheads are substituted methyl ketones, aldehydes, epoxides, aziridines, haloketones, and Michael acceptors. Over the years, inhibitors have been developed with remarkable selectivity for various cysteine proteases. 

Crystallographic structures are available for both *Homo sapiens* ATG4A and ATG4B [69,70,71]. As discussed above, ATG4A and ATG4B show slightly different substrate preference, with ATG4A preferring GABARAPL2 compared to LC3 family members, at least in vitro. Both ATG4A and ATG4B share high sequence and structural homology, and the only notable residue that is involved in LC3 binding that is absent in ATG4A is Leu232, which is the more steric Ile233 in ATG4A at the corresponding position. It has been observed that substitution of Ile233Leu resulted in an ATG4A variant that is able to cleave LC3B, thus rendering it more similar to ATG4B [70]. In its free form, both ATG4A and ATG4B are auto-inhibited by the regulatory N-terminal loop that folds over the catalytic Cys74, thus shielding it completely. This inhibitory loop is displaced upon binding to LC3/GABARAP and, in addition, is most likely subject to conformational change induced by post-translational modification. For instance, Ser34, which is located at the hinge of the N-terminal domain, has been shown to be phosphorylated by AKT kinase [14], and a phospho-mimetic S34D mutant displays enhanced ATG4B activity [72], suggesting that phosphorylation results in a conformational opening of the auto-inhibitory loop. 

In addition to the catalytic site, two other sites for binding to the LC3/GABARAP substrate have been identified: A C-terminal LC3 Interacting Region (LIR) and an N-terminal LIR domain [60,73,74]. The N-terminal LIR motif is required for activation of ATG4B upon LC3 binding: Mechanistically, it is thought that Phe119 in LC3 causes a large conformational shift in the N-terminal region, allowing entry of the LC3 tail into the active site. The N-terminal tail containing the LIR sequence Tyr–Asp–Thr–Leu then binds to another (free) LC3 molecule, thus enabling the entry of the substrate sequence. This model supports the notion that full-length LC3 is required for efficient activation of ATG4B and it has been observed that LC3 peptides are less efficient in doing so [75]. The C-terminal LIR region is important for catalytic activity as well, and, in addition, regulates protein stability of the substrates [73].

## 3. ATG4B in Cancer

ATG4B has been implicated in several diseases, including pulmonary fibrosis [76], lung endotoxemia/sepsis [77], colitis [78], ischemia/reperfusion [79], Huntington’s disease [80], and various forms of cancer. 

### 3.1. Breast Cancer

ATG4B expression is highly elevated in human epidermal growth factor receptor 2 (HER2)-positive breast cancer [81]. In this study, it was shown that HER2-positive breast cancer cells, but not HER2-negative cells, required ATG4B for survival under stress conditions. Further, siRNA-mediated knockdown of ATG4B and HER2 resulted in a decrease of breast cancer cell viability, and a combination of ATG4B knockdown with the HER2 inhibitor trastuzumab resulted in a strong reduction of cell viability in HER2-overexpressing cells [81]. On the other hand, suppression of ATG4B using a dominant negative version resulted in increased cell migration and potential for increased lung metastases in a model of hypoxic breast cancer cells [82]. Other findings report that treatment of triple negative breast cancer (TNBC) cells with the ATG4B agonist Flubendazole resulted in an induction of autophagy and reduced TNBC growth [83]. Thus, in the case of breast cancer, evidence suggests that modulation of ATG4B provides a benefit in pre-clinical models. Further studies are required to determine the suitability of ATG4B inhibitors or agonists for therapeutic strategies.

### 3.2. Pancreatic Cancer

There is strong evidence that pancreatic cancer cells are particularly vulnerable to inhibition of ATG4B. This is particularly evident in pancreatic ductal adenocarcinoma (PDAC) that is dependent on mutant K-Ras expression [84]. The mechanism of how autophagy contributes to PDAC growth is not completely clear, and various models have been suggested, including the provision of metabolites and the ability of the tumour to cope with stresses [85]. Recently, it was proposed that Ribose-5-phosphate isomerase (RPIA), a regulator of LC3 processing and autophagy [65], contributes to resistance of KRas-dependent pancreatic cancer via the regulation of nucleotide synthesis [86]. Autophagy inhibition in K-Ras-dependent PDAC also resulted in increased DNA damage and apoptosis, corroborating the idea that autophagy limits DNA damage stress in PDAC tumours, which may be affected by p53 status [87,88]. Autophagy inhibition may also affect the surrounding stromal cells, as has been shown recently [89]. In this study, it was shown that autophagy in pancreatic stellate cells generate secreted alanine, which is required for the tumour to grow. In support of this hypothesis, data are emerging that inhibition of autophagy by administration of the drug hydroxychloroquine in combination with chemotherapy can be beneficial to reduce tumour growth [90]. Recently, a very elegant study was performed to assess the consequence of ATG4B inhibition in already-formed tumours, using a genetically engineered mouse model expressing the catalytic inactive version of ATG4B [36]. In this model, expression of the ATG4B mutant is doxicyclin-dependent, and thus treatment of mice with doxicyclin with different dosing regimens faithfully reflects the dosing regime of potential drugs in the clinic. Several key findings were made that provide strong evidence that targeting ATG4B in PDAC is a potential anti-cancer strategy. First, it was observed that tumours were regressing as early as after 4 days of doxicyclin treatment and maintained over 2 weeks of treatment. Similar results were seen in a K-Ras mutant lung cancer genetically engineered mouse model (GEMM) expressing dominant negative ATG4B [36]. However, in the PDAC model, it was puzzling that there was no significant increase in survival of the animals, which was due to disruption of pancreatic function in the non-tumour cells. This potential organ toxicity following autophagy inhibition in non-tumour cells was dependent on mutant K-Ras expression, supporting the notion that additional oncogenic insults may be required for adverse effects on organ health by autophagy inhibition. One possibility to overcome this caveat was by using intermittent, reversible administration of doxicyclin in this model, which resulted in strong reduction of tumour growth and extended animal survival [36]. Furthermore, it was observed that nutrient-deprived cells both ex vivo and in vivo were particularly vulnerable to doxicyclin treatment and showed increased apoptosis, suggesting that the stresses that tumour cells encounter predispose these to such treatments. 

### 3.3. Lung Cancer

There is strong evidence that lung cancer is particularly vulnerable to inhibition of autophagy [8]. This stems from the observation that autophagy is highly up-regulated in lung cancer, and inhibition of autophagy by loss of ATG7 expression can suppress tumour cell growth in vitro and in xenograft models [8]. This is particularly evident for Ras-driven lung cancers, where ATG7 deletion results in increased mitochondrial mutation load, impaired mitochondrial function, increased production of reactive oxygen species, and a reduction in the cellular nucleotide pool [91]. In a genetically-engineered mouse model of K-Ras-driven lung cancer, the expression of doxycycline-inducible dominant negative ATG4B C74A resulted in a strong reduction in lung cancer cell growth [36], suggesting that inhibition of ATG4B is a promising strategy for treatment of lung cancer.

### 3.4. Colorectal Cancer

ATG4B expression was found elevated in colorectal cancer, and knockdown of ATG4B resulted in reduced cell cycle progression and inhibition of colorectal cancer cell lines [92]. Some ATG4B inhibitors identified to date have been tested in xenograft models of colorectal cancer cells, typically HCT-116 cells, and demonstrated a significant benefit in reducing tumour growth, in particular when combined with chemotherapeutic agents [93]. 

### 3.5. Prostate Cancer

ATG4B is an androgen receptor responsive gene that correlates with disease progression in prostate cancer [94]. In this study, it was suggested that higher levels of ATG4B are essential for maximal growth of prostate cancer cells. Another study showed that siRNA-mediated knockdown of ATG4B in prostate cancer cells resulted in increased apoptosis, further supporting the notion that inhibition of ATG4B may be beneficial for reducing prostate cancer growth [95]. This is in contrast to another study that suggested miRNA34A-mediated chemoresistance in prostate cancer cells is—at least in part—mediated by reduced ATG4B expression. The authors proposed that down-regulating ATG4B expression through miRNA34A made prostate cancer cells more prone to chemoresistance and survival under stress [96]. Another study suggested that the role of ATG4B in prostate cancer is cell type-, treatment-, and context-dependent [97], showing that the use of a dominant negative construct can either enhance the effect of chemotherapy or contribute to treatment resistance. To circumvent this problem, the identification of predictive markers of response was suggested. 

### 3.6. Other Cancers

In a study investigating chronic myeloid leukaemia cells, it was reported that ATG4B, ATG5, and Beclin-1 expression levels are significantly increased [53]. In particular, it was found that knockdown of ATG4B by siRNA suppresses growth and survival of CD34+ chronic myelogenous leukemia (CML) progenitor cells and sensitises them to treatment with imatinib mesylate. The authors further suggested that expression levels of ATG4B are predictive of treatment responses and could serve as a biomarker for CML. In osteosarcoma, the use of ATG4B antagonists provided a benefit in reducing growth of osteosarcoma tumours [98]. A role for autophagy in glioblastoma multiforme (GBM) has been proposed, and it was shown that inhibition of ATG4B in combination with radiotherapy can slow tumour growth in intracranial GBM xenograft models [15]. 

## 4. Screening Assays and Tool Compounds

### 4.1. Screening Assays

Several biochemical and cellular assays that specifically measure ATG4 protease activity to develop small molecule inhibitors for ATG4B are available. These assays can be broadly divided into biochemical assays, measuring directly the activity of ATG4; computational methods for docking studies; and cell-based assays, monitoring ATG4B activity through an indirect readout—often reporter assays. Biochemical assays tend to be preferred in a drug discovery setting for two main reasons: One is that these assays are typically easily automated and relatively cheap, thus enabling high-throughput screening campaigns. The other key advantage is that the compounds typically act directly on the target protein, thus enabling studies of structure–activity relationship (SAR) and providing a much better understanding of how the compound engages with the target. On the other hand, biochemical assays do not give any information about toxicity or cell penetration, which is the most important advantage of cell-based assays. Most importantly, cell-based assays allow the characterisation of direct functional readouts. Typically, a large-scale biochemical screening campaign will be followed by secondary assays, including cell-based assays. 

#### 4.1.1. Biochemical Assays

Biochemical assays to measure the ATG4B protease activity are available (Figure 3).

These range from simple fluorometric readouts [99], fluorescence resonance energy transfer (FRET)-based sensors [99,100] to gel electrophoresis assays [101]. Typically, assays to monitor ATG4B activity rely on the cleavage of its substrate LC3A or LC3B that can be monitored by a coupled enzymatic or fluorescent reaction. Small fluorescent peptides have been synthesised to monitor ATG4B activity in a simple fluorescence readout [99]. In this case, the most effective substrate combined short peptides from the C-terminus of LC3B, such as FG and TFG linked to 7-amino-4-carboxamidomethyl-coumarin (ACC). Longer peptides were less efficient. The turnover was further enhanced by the addition of N-capping chains such as HO_2_C(CH_2_)_5_CO–NH and an improved linker. In a proof-of-concept screen using the library of pharmacologically active compounds (LOPAC) library of Federal Drug Administration (FDA)-approved drugs, this substrate was suitable for large-scale screening and identification of ATG4B inhibitors. 

Alternative biochemical assays utilise the fusion of longer peptides or proteins at the C-terminus of LC3 that are released by ATG4B-mediated cleavage and can result in a change of molecular weight (e.g., LC3-glutathione-S-transferase (GST), LC3-red fluorescent protein (RFP)), or release of enzymatic activity such as phospho-lipase A2 (PLA2) [75]. The LC3–GST assay is typically used in small-scale experiments and is a direct readout of ATG4B in vitro activity. Recombinant ATG4B and LC3–GST are mixed together in an appropriate buffer and incubated for a limited time at a permissive temperature, whereupon LC3–GST is cleaved into the LC3 and GST entities that can be resolved by poly-acrylamide gel electrophoresis and Coomassie Brillant Blue or Oriole fluorescent gel staining [102]. Both components have to be carefully titrated and the assay conditions optimised in order to be able to accurately determine protease activity kinetics. These assays have been successfully employed to identify small molecule inhibitors and agonists [103], but have some limitations in terms of sensitivity and ease of use. The PLA2 assay couples cleavage with release of the enzyme PLA2, which can then be monitored by conversion of a fluorescent substrate [103]. Another in vitro assay makes use of fluorescence resonance energy transfer (FRET), whereby LC3 is flanked with a fluorescent acceptor molecule and a fluorescent donor molecule. Combinations of cyan fluorescent protein and yellow fluorescent protein (CFP–LC3–YFP) [104], and YFP and EmGFP (YFP–LC3B–EmGFP) [100] have been used, as well as LC3B and GATE-16 peptides where an N-terminal His-tag and a C-terminal GST-tag were labelled with Eu and Ulight, respectively [105]. While the construct results in strong fluorescence energy transfer in uncleaved state, the cleavage results in loss of such energy transfer and, thus, an increase in the donor emission. This assay has been used both in vitro and in cells. Also, the assay can be confirmed by Western blotting due to the reduced size of the FRET construct after cleavage. Generally, FRET-based assays require careful optimisation, and common issues can arise from spectral overlap of the donor and acceptor molecule and makes quantification difficult. Thermal shift assays have also been tested to identify binders of ATG4B, but was not further pursued for large-scale screening due to the high amount of enzyme consumption and variability [102]. 

#### 4.1.2. Computational Assays

An alternative to biochemical assays is the use of computational docking for identification of binders to functional domains in ATG4B. There are several pockets on the surface of ATG4B that are suitable for docking studies, including the active site, LIR domains, or other pockets required for function [60]. The crystal structure for ATG4B in complex with its substrate LC3B is available, thus facilitating such studies. It should be noted that the C-terminal LIR is not well resolved by crystallisation to date, and it has not yet been fully demonstrated whether interfering with this domain has therapeutic relevance. Many effective agonists and antagonists for ATG4B have been identified, some through a combination of molecular docking studies and experimental validation. For instance, NSC185058, an agonist with moderate potency, was first identified in silico and validated using cell-based assays [98]. More recently, computational docking studies have identified styrylquinolines as binders to ATG4B, and have subsequently been shown to reduce activity [106]. Another potent inhibitor called S130 was identified by a similar in silico approach and validated with the LC3B–GST assay [101]. Computational docking studies have also resulted in the identification of an agonist for ATG4B, Flubendazole [83]. Thus, such computational docking for the identification of inhibitors is a promising approach to identify small molecule binders for ATG4B. Furthermore, the identification of ATG4B binders may enable the development of proteolysis-targeting chimeras (PROTACs) [107], a promising alternative to the development of ATG4B inhibitors. 

#### 4.1.3. Cell-Based Assays

Cell-based assays use reporter genes such as luciferase or fluorescent proteins to visualise the consequence of ATG4B inhibition or activation (Figure 3). This can be achieved by tagging of the substrate LC3B with either a fluorescent or luminescent reporter gene and monitoring changes in fluorescence or luminescence. A dual tagged GFP–LC3–RFP, for instance, will result in GFP-only autophagosomes upon cleavage of the C-terminus by ATG4B. FRET-based assays, as explained above, where a fluorescent donor protein (YFP) and a fluorescent acceptor protein (CFP) are fused at the N- and C-terminus of LC3B/GATE16, respectively, have been generated and used in medium-throughput screening [104]. A two-component FRET-based assay, in combination with Fluorescence recovery after photo-bleaching, was developed to investigate interactions or ATG4B with LC3 [108] with potential for large-scale phenotypic screening, but with limited application to investigate the cellular activity of ATG4B. To overcome this problem, more selective assays employing luciferases have been developed. One example is the luciferase release assay, where LC3B is tagged at the C-terminus with Gaussia luciferase (GLUC) [109,110]. ATG4B-mediated cleavage results in release of GLUC by non-conventional secretion [111] that can be non-invasively monitored by harvesting supernatants from cell culture. The mechanism of GLUC release in this system is not well understood, but it does not require autophagy as it is not affected in ATG5 knockout cells [111], making this a specific ATG4B activity assay. This assay has been employed to identify regulators of ATG4B activity and has become one of the main methods to monitor cell-based ATG4 activity [12,59,72]. The assay has also been adapted to monitor LC3A and LC3B2 [9], thus broadening its application to other ATG4 family members, although a similar approach to monitor GABARAP cleavage instead of LC3 resulted in very high background release of GLUC, and is thus impractical for use as a phenotypic reporter for screening (R.K., unpublished observation). Another luciferase-based sensor utilises the complementation of a split luciferase that is flanking LC3. In this arrangement, the uncleaved LC3 displays high levels of luminescence that is lost when cleaved by ATG4B [93]. This assay has been used to validate ATG4B inhibitors in cells, but the dynamic range of this assay is limited due to a high background activity. An assay to monitor transcriptional activation of ATG4B by fusion of the ATG4B promoter with renilla luciferase has also been reported [72], which may be useful for investigating upstream regulators of ATG4B expression. For instance, this assay has been used to identify small molecule compounds that activate ATG4B expression [72]. A common drawback to most luciferase-based reporter assays is that it is sensitive to changes in cell viability and transcription/translation. Thus, the use of a control luciferase may be required to normalise for such effects. One advantage of the method is that it may be used to monitor in vivo activity of ATG4B, since GLUC is highly stable and can be measured ex vivo from serum [112]. 

### 4.2. Tool Compounds

ATG4B has been proposed as a drug target elsewhere [113,114]. There is increasing evidence that modulation of ATG4B by either si/shRNA-mediated knockdown or the expression of a dominant negative construct results in a benefit in multiple cancer models, including breast, pancreatic and lung cancer (Table 1). 

Several agonists and inhibitors of ATG4B have been described in the literature and reviewed [60]. These were identified either by structure-guided molecular docking of compounds in silico, or by screening chemical libraries of compounds with known activity, such as FDA-approved drugs (Table 2). 

#### 4.2.1. ATG4B Inhibitors

A variety of ATG4B inhibitors have been described with varying chemical scaffolds (Table 2). NSC185058 directly binds to and inhibits ATG4B in vitro and reduces growth of SAOS osteosarcoma cells in vivo [98]. NSC185058 interacts in the pocket containing His280 and Asp278, which are part of the catalytic triad. Treatment of cells with this compound inhibits autophagy in an mTOR-independent manner. However, the inhibitory activity is not very potent and selectivity of this compound towards other cellular targets has not been assessed. 

Using a FRET-based assay, two other inhibitors for ATG4B were identified, hypericin and aurin tricarboxylic acid with IC_50_ values of 57 μM and 4.4 μM, respectively [100]. In this study, an YFP–LC3–EmGFP reporter was used in a screen of ~5000 compounds of bioactive compounds and known drugs. The primary screen was done using the fluorescent FRET signals, and cleavage of the reporter was confirmed by mass spectrometry. Hypercin, aurin tricarboxylic acid, and Z-L-Phe-chloromethylketone were identified with IC_50_ values of 57 μM, 4.4 μM, and 99 nM, respectively. 

Using the LC3–GST assay and Western blotting, a fragment library of 182 fragments was screened to identify a fragment termed “S069” as inhibitor for ATG4B [102]. Based on these findings, benzotropolone derivatives were synthesised and shown to inhibit LC3 processing in Jurkat T cells [115]. One of these compounds, UAMC-2526, was tested in the GFP–LC3 transgenic reporter mouse model, and showed that subcutaneous delivery of UAMC-2526 blocked autophagy and LC3 processing in the liver. Furthermore, UAMC-2526 reduced colorectal cancer cell growth in vitro and significantly reduced colorectal xenograft tumour growth in combination with the chemotherapy drug Oxaliplatin. Inhibition of autophagy in the xenograft tumours by this combination treatment was confirmed by electron microscopy. On-going studies aim to further optimise benzotropolone derivatives as potent ATG4B inhibitors [119]. 

Tioconazole was identified as an ATG4A and ATG4B inhibitor using a combination of in silico screening of a collection of FDA-approved drugs and validation with biochemical assays, notably a GABARAPL2-PLA assay and a split-luciferase reporter [93]. Tioconazole was found to dock to the active site of ATG4B, and also to a second site that is required for binding of LC3 at the N-terminus. In cancer cells, tioconazole reduced autophagic flux and decreased the size of HCT-116 spheroids and HCT-116 xenograft tumours, which correlated with impaired LC3 processing in tumour tissues. 

Another study using computational docking for identification of ATG4B binding molecules presents results from an in silico screen using a library of 7249 diverse compounds, and identified S130 as a small compound that binds to a pocket in proximity to the active site of ATG4B [101]. S130 results in the accumulation of lipidated LC3 in cells, indicating disruption of autophagic flux, and has strong anti-proliferative activity in a variety of cancer cell lines, as well as in vivo growth of colorectal cancer xenografts [101]. Furthermore, a LC/MS-based assay to monitor the pharmacokinetic properties of S130 has been recently developed [120].

Using molecular docking of a larger collection of >730,000 compounds, styryl-quinolines were identified as strong binders of ATG4B [106]. Compounds were validated by mass spectrometry of the GFP–LC3–YFP reporter and a fluorescent peptide, demonstrating that compound LV-320 inhibits LC3 processing and autophagy in cancer cells and in the liver of GFP–LC3 transgenic mice in vivo. LV-320 is a uncompetitive allosteric inhibitor of ATG4A and ATG4B, but did not inhibit other proteases such as caspase-3 or cathepsin B. 

A recent high-throughput virtual screening approach exploring the NCI Open Database of 265,242 compounds identified benzo[cd]indol-2(1 H)-ones as binders for ATG4B and validated some hit compounds using an alphascreen and mass spectrometry assay [118]. Through modifications, compounds with micromolar IC_50_ values were identified in this campaign. 

7-keto-cholesterol (7-KC) was also identified as an inhibitor of ATG4B using the LC3–PLA2 assay, which is accompanied by effects on cellular redox mechanisms [116]. Interestingly, 7-KC increases autophagic flux in smooth muscle cells, which is most likely mediated by reactive oxygen species. 

Very potent inhibitors based on chloromethyl- or fluoromethyl-ketone warheads that mimic the peptide substrate were identified using a FRET-based assay [105,117]. The methyl-ketone forms a covalent linkage to the active site cysteine, thus inhibiting activity. Further structure-guided optimisation resulted in FMK-based peptidomimetics with higher potency to achieve IC_50_ values in the lower nanomolar range. However, the FMK-based inhibitors show some inhibitory activity towards other cysteine proteases with IC_50_ values < 1 μM as well [117]. Interestingly, the most potent ATG4B inhibitor to date, FMK9A, also induces autophagy through a mechanism that is independent of ATG4B activity [121], thus limiting its application as a tool compound to monitor ATG4B-specific effects. 

It is interesting to note that ATG4B inhibitors were identified in natural product sources as well, such as Xanthium strumarium food extract, though the compound has not yet been purified [122].

#### 4.2.2. ATG4B Activators

ATG4B agonists have also been proposed to be beneficial for anti-cancer strategies. For instance, *N*-Benzoyl-*O*-(N’-(1-benzyloxycarbonyl-4-piperidiylcarbonyl)-d-phenylalanyl)-d-phenyl-alaninol (BBP) is an agonist for ATG4B that induces autophagy in MCF-7 cells by increasing the expression of ATG4B [123]. Prolonged treatment of MCF-7 cells with this compound results in autophagic cell death. Another agonist, Flubendazole, that was identified by a combination of in silico docking and experimental validation also induces autophagic cell death upon prolonged times of treatment and has been proposed to be of benefit for the treatment of TNBC [83]. However, both compounds have only been tested in cell lines so far and lack the potency and selectivity to advance to further drug development. Overall, the consequence of autophagic cell death and the exact mechanisms employed by these compounds to include autophagic cell death require further investigation. Another compound, STK683964, induced cellular ATG4B activity through an unknown mechanism that involves up-regulation of ATG4B expression [72]. 

#### 4.2.3. Conclusion

Overall, most known activators and inhibitors of ATG4B are useful tool compounds for setting up screening cascades, but they lack the potency and, in some cases, the characteristics as useful lead compounds for drug development. Continued efforts are required to further identify and optimise selective and potent ATG4B inhibitors. In summary, the development of potent ATG4B inhibitors is possible, though selectivity of such compounds has not been fully assessed to date. 

## 5. Biomarkers, Assays for In Vivo Activity, and Target Engagement

In the past, one of the major hurdles in the development of therapeutic agents targeting autophagy is the absence of robust biomarkers and assays to monitor autophagy in vivo. There is considerable progress in both areas, which may enable better assessment of small molecule compounds in pre-clinical models in the near future. Some in vivo models used for inhibitor studies and biomarkers are summarised in Table 1. Biomarkers are generally classified as (a) pharmacodynamic that allow measuring target engagement (e.g., in the case of ATG4B inhibition, the lack of LC3 lipidation and p62 accumulation; (b) predictive that allow predicting whether the tumour will respond (e.g., high ATG4B expression as proposed in stem cells/CML and breast cancer); and (c) prognostic that tell something about the disease outcome (tumour growth) regardless of therapy (e.g., HER2-positive breast cancer or K-Ras mutant PDAC). Pharmacodynamic biomarkers could include proteins of the autophagy pathway such as LC3/GABARAP, p62, and NDP52. One caveat with these is that the expression level or level in serum does not necessarily correlate with any autophagic activity in vivo. There are very few proteins that are specifically induced upon autophagy and absent under basal conditions. This is further complicated by the fact that basal autophagy is active in all cells under steady-state conditions. There are several autophagy mouse models which have provided insights into the dynamic regulation of autophagy in tissues in the context of a whole animal and which might help to identify candidate pharmacodynamic biomarkers in the future [124]. 

As mentioned above, there is clear evidence that targeting autophagy in cancer can be beneficial. One of the biggest problems is that a clear definition of autophagy in the various stages of a cancer is missing. The general concept predicts that a growing solid tumour will go through a phase of nutrient limitation, where autophagy activity is predicted to be high, thus enabling the tumour to survive this period of nutrient stress. Formally, this has not been experimentally validated. Good animal models to monitor autophagy in vivo are required to determine the in vivo vulnerability of growing tumours to autophagy modulation. 

Several mouse models exist, such as the GFP–LC3 transgenic mouse where GFP–LC3 expression is driven by the constitutive CAG promotor (cytomegalovirus immediate early enhancer and chicken (β−actin promoter), abundantly in all tissues [125]. This model is frequently used to monitor LC3 levels at steady-state in the liver. Since ATG4B inhibition is expected to result in a block of pro-LC3 processing, the absence of LC3 puncta in these tissues can be interpreted as a block in LC3 processing. However, one needs to be careful, since other mechanisms might result in a loss or reduction of LC3 expression, making interpretation difficult. Also, fluorescent protein aggregates such as lipofuscin complicate interpretation of results using this model. Furthermore, GFP fluorescence is quenched in the low pH of the lysosome. Therefore, a tandem RFP–GFP–LC3 transgenic mouse where RFP–GFP–LC3 is under the control of the CAG promoter has been generated [126]. A transgenic mouse expressing the mCherry–GFP–LC3 tandem reporter has been generated, enabling in vivo analysis of autophagy [127]. Recently, a modified reporter for this system has been presented, where GFP was exchanged for pHluorin, a more sensitive pH indicator [128]. A variant of this is the mito-QC reporter mouse, where the tandem RFP–GFP reporter is targeted to mitochondria by the mitochondrial targeting sequence of the outer mitochondrial membrane protein FIS1, thus enabling the in vivo monitoring of mitophagy [127,129]. While this model enables to distinguish autophagosomes (green) and auto-lysosomes (red), it still fails to give an accurate readout of the dynamic turnover of autophagosomes and autophagic flux. Also, variations in reporter gene expression in tissues is not taken into account, making it difficult to assess autophagy or autophagosome turnover in tissues. A control reference for normalisation to the overall expression level of the reporter gene is missing. The GFP–LC3–RFP–LC3ΔG reporter can overcome this problem [130], whereby ATG4B cleaves the reporter into autophagosome-associated GFP–LC3 and cytoplasmic RFP–LC3ΔG. The amount of cytoplasmic RFP can be used to normalise and quantify autophagic flux. While this is a big step forward to facilitate a better understanding of autophagy in vivo, some of the caveats mentioned above still remain, such as endogenous fluorescence, quenching of GFP in the lysosome, and limitations to measure directly and specifically ATG4B activity. Furthermore, all models generated to date require ex vivo measurements of autophagy. A non-invasive real-time assessment of autophagy activity would be required and could be potentially facilitated by the use of monitoring of luciferase activity in serum, such as described for in vivo proliferation assays [112], and monitoring caspase-1 protease activity non-invasively in mice [131]. 

Another important aspect of evaluating compounds in vivo is the ability to measure target engagement in various tissues. Typically, binding of compounds to other cysteine proteases can be achieved by the use of warheads that are displaced upon binding of the compound to the active site. For instance, engagement of de-ubiquitinating enzymes with a small molecule inhibitor can be validated by the use of a non-hydrolysable ubiquitin and competition for active site binding with the compound. No such target engagement assay has been described for ATG4B yet, and efforts are required to optimise such assays. 

## 6. Conclusions 

In conclusion, we propose that human ATG4B is a druggable target with strong potential for the development of potent inhibitors. However, several unknowns still remain that will need to be addressed in the future. Overall, there is clear evidence that targeting ATG4B can provide a benefit in cancer, and there is particularly good evidence for breast cancer, lung cancer, colorectal cancer, and pancreatic ductal adenocarcinoma. Another important aspect is how ATG4B inhibition may affect host cell autophagy that has been shown to fuel tumour growth through various mechanisms, such as arginine secretion [11]. The biology of ATG4B and its role in autophagy is quite well understood, although some uncertainties arise from the redundancies of the ATG4 family of proteins. For instance, it is not yet clear whether selectivity for ATG4B inhibitors over ATG4A is feasible and whether this actually matters, given that the contribution of ATG4A to basal autophagy is negligible. Further studies looking at double knockout ATG4A/B mice will be required to address this. A big advantage for a drug discovery campaign is that common tools that enable drug discovery are established, including the availability of a crystal structure, screening assays, and tool compounds (though not very potent at this stage). Clearly, more potent compounds will be required for studies in pre-clinical animal models, but given the availability of high-throughput screening assays, this should be feasible. One caveat is the absence of translational biomarkers, and efforts to identify good autophagy biomarkers should be prioritised. This could take the form of expression signatures, or the measurement of potential ATG4B substrate levels in cells. However, the use of LC3/GABARAP as biomarkers proves difficult, given that their abundance is regulated at multiple levels. Further downstream substrates might be more reasonable, such as the levels of autophagy cargo proteins that may reflect on the activity of the overall process rather than directly on ATG4B activity. 

## Figures and Tables

**Figure 1 cells-09-00053-f001:**
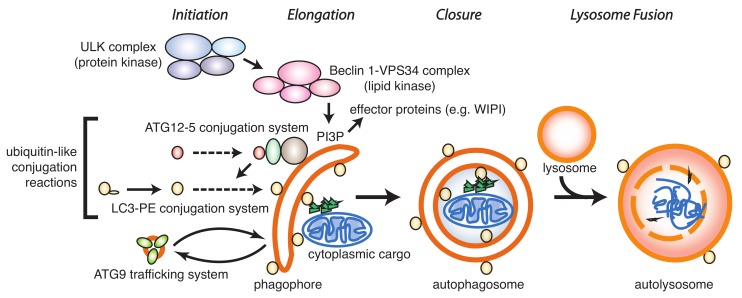
Key components in the autophagy pathway. Autophagy is initiated by the ULK protein kinase complex, leading to activation of the Beclin-1–VPS34 complex and generation of phosphatidyl-inositol-3-phosphate (PI3P). PI3P serves as a recognition domain in the phagophore membrane for recruitment and assembly of other protein complexes such as the WD-repeat protein interacting with phosphoinositides (WIPI) proteins. These early events lead to activation of a series of ubiquitin-like conjugation reactions, including ATG12-5 formation and LC3/GABARAP processing. Recruitment of cargo to the phagophore and closure of the autophagosome is followed by fusion with the lysosome and degradation of cargo in the autolysosome.

**Figure 2 cells-09-00053-f002:**
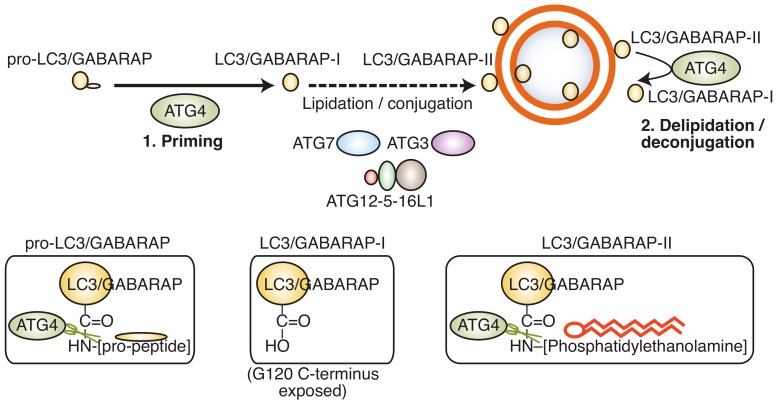
LC3/GABARAP protein processing. The LC3/GABARAP pro-peptide pro-LC3/GABARAP is first cleaved at the C-terminus by ATG4B, generating LC3/GABARAP-I in the cytoplasm. LC3/GABARAP-I is then conjugated to phosphatidylehtanolamine (PE) by ATG7/ATG3 and the ATG12-5–16L1 complex. LC3/GABARAP-PE can be deconjugated by ATG4 proteins for recycling. The different C-terminal ends of pro-LC3/GABARAP, LC3/GABARAP-I, and LC3/GABARAP-II are depicted in the bottom panels.

**Figure 3 cells-09-00053-f003:**
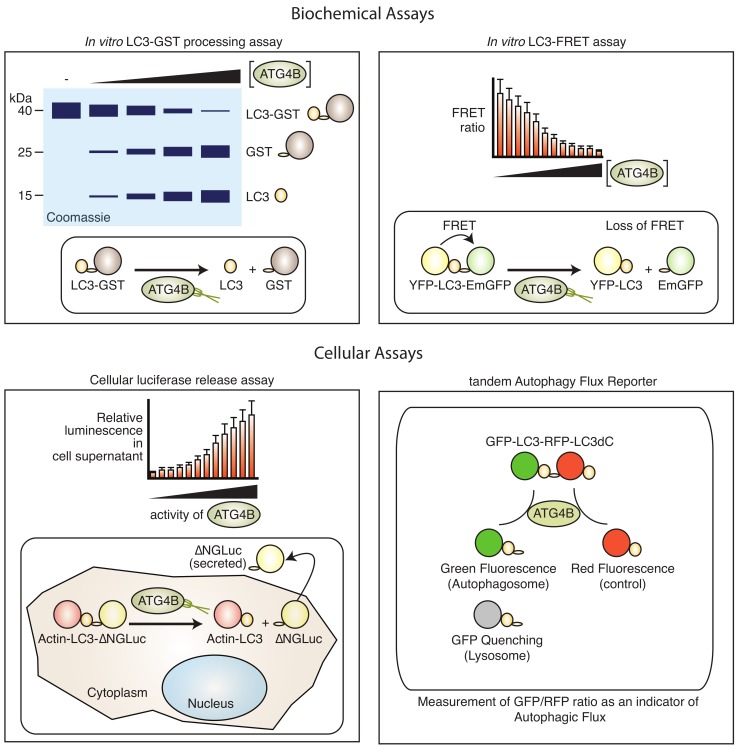
Biochemical and cell-based assays to monitor ATG4B activity. Biochemical assays such as the in vitro GST–LC3 processing assay (top left) that relies on poly-acrylamide gel electrophoresis are often used to monitor cleavage of a LC3–GST pseudo-substrate, but are not suitable for high-throughput screening applications. The LC3–FRET assay (top right) has been used in large-scale screening applications in vitro. The luciferase release assay (bottom left) that relies on ATG4B-dependent release of an N-termially truncated Gaussia luciferase (ΔNGLuc) enzyme and export into supernatants of cells is suitable for cell-based high-throughput screening. The tandem autophagy flux reporter (bottom right) that monitors ATG4B-dependent cleavage of a tandem GFP–LC3–RFP–LC3dC (C-terminal truncation) is suitable for cell-based high-content screening and in vivo. For more detail on each assay see text.

**Table 1 cells-09-00053-t001:** In vivo models and potential biomarkers for ATG4B inhibition in cancer.

Cancer Type	Therapeutic Modality	In Vivo Model	Biomarker	Reference
Breast cancer	siRNA ATG4B/Trastuzumab	MCF7 xenograft	HER2, ATG4B	[81]
Colorectal cancer	Tioconazole	HCT-116 Xenograft	none	[93]
Colorectal cancer	S130/ Caloric restriction	HCT-116 Xenograft	none	[101]
Colorectal cancer	UAMC2526/ oxaliplatin	HT-29 Xenograft	LC3 conversion	[115]
Glioblastoma	NSC185058/ Chloroquine	M83 glioma xenograft	none	[15]
Lung adenocarcinoma	Doxicylcin-inducible ATG4B C74A	GEMM	K-Ras mutation	[36]
Osteosarcoma	NSC185058/ starvation	SAOS Xenograft	none	[98]
Pancreatic ductal adenocarcinoma	Doxicyclin-inducible ATG4B C74A	GEMM	K-Ras mutation	[36]
Prostate cancer	ATG4B C74A/ doxorubicin	PC-3 Xenograft	none	[97]

**Table 2 cells-09-00053-t002:** ATG4B inhibitors and their effect on cancer cell growth. N.D., not described.

Compound	Chemical Scaffold	Screening Assay	Cell Type	Cancer Type	Reference
7-keto-cholesterol	Keto-cholesterol	LC3–PLA2	HASMC	N.D.	[116]
Aurin-tricarboxylic acid	Polyphenole	FRET (YFP–LC3–EmGFP)	N.D.	N.D.	[100]
FMK9A	Methyl-ketone	TR–FRET	N.D.	N.D.	[117]
Hypericin	Anthra-quinone	FRET (YFP–LC3–EmGFP)	N.D.	N.D.	[100]
LV-320	Styryl-quinoline	In silico	SKBR3, MCF7, JIMT1, MDA-MB-231	Breast Cancer	[106]
NSC185058	Pyridine-carbothioamide	In silico	SAOS M83	Osteosarcoma Glioblastoma	[98] [15]
NSC611216	Benzo-indolone	Alphascreen	HT-29	Colorectal cancer	[118]
S130	Dibenzo-quinoline	In silico	HCT-116	Colorectal cancer	[101]
Tioconazole	Dichlorphenylethyl-imidazole	In silico, GABARPL2–PLA	HCT-116	Colorectal cancer	[93]
UAMC2526	Benzo-tropolone	LC3–GST	HT-29	Colorectal cancer	[115]

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
