# Peer review of "On ATG4B as Drug Target for Treatment of Solid Tumours—The Knowns and the Unknowns"

_cells, 2019, doi:10.3390/cells9010053_

Round 1
Reviewer 1 Report
Comments to the Author:
The authors have focused on human ATG4B which is a druggable target with strong potential for the development of potent inhibitors to target different cancer. However, the reviewer feels that there are minor changes in the review that needs to be addressed. The reviewer feels that the addition of a few literature and tables will enhance the outlook of this review. Please find the suggestions below.
The author says that autophagy is lethal during the initiation of cancer and acts as survival during its progression. How the expression of ATG4B is altered during these two phases concerning different cancer? Please include the role of ATG4B in fibrosarcoma under “other cancer” What are the advantages of using ATG4B as a druggable target compared to its other variants, not only ATG4A? Line 193-199, the author talks about the auto-inhibitory loop of ATG4B and ATG4A. Please provide examples of this phenomenon in any cancer. It is highly recommended to draw tables. Table 1. for clinical aspects, depicting the screening assays and tool compounds targeted for different cancer or animal model. Table 2. for biomarkers, assays for in vivo activity and target engagement for different types of cancer used for targeting ATG4B. ATG4B is a potential biomarker and therapeutic target in stem/progenitor cells. Please include this in the manuscript.Author Response
We thank the reviewer for these very useful comments. Following the advice, we have included two new tables. Table 1 summarises the tool compounds and screening assays in various cancer types. Table 2 summarises biomarkers and in vivo reports, where inhibition of ATG4B has been studied.
We have also included a statement on ATG4B as a potential biomarker in stem/progenitor cells.
Regarding the comment how the expression of ATG4B is altered in different stages of cancer: this is an important and interesting point but so far there is no report in the literature to address this question.
Reviewer 2 Report
A. Agrotis and colleagues have provided a complete review about the use of ATG4B as a druggable cancer target. Nevertheless, few comments should be addressed to clarify certain aspects and strengthen the main message of this review:
1.- In the “Introduction” section, authors explain the different steps of autophagy pathway, and where LC3 appears and contributes to it. In Figure 1, authors should mention the distinct stages of this degradative pathway as well. Related to that, in line 69, they highlighted that LC3 links the autophagosome membrane with the machinery. This statement is partially true, since LC3 is also present during the phagophore elongation, and not only after the autophagosome closure.
2.- In Figure 2, authors dissect how LC3 is processed by ATG4B. They should correct the nomenclature, because LC3B isoform is not the only isoform conjugated/deconjugated via this system, but the whole LC3/GABARAP family, as nicely described in the text of the manuscript. Moreover, in the same figure, it is mentioned LC3B-PE and LC3B-II below as there are 2 different conjugated-LC3B forms, providing some confusions to the reader.
3.- In the part 3 of the review, authors deeply describe the role of ATG4B in several forms of cancer. Since ATG4B has a different role depending where the cancer is located, it would be nice that they prepare an extra figure showing the distribution of the explained cancer and how ATG4B is relevant in each of them.
4.- Authors should improve the section 4 of the review (Screening Assays and Tool Compounds). It is suggested that they define a subsection 4.1. for Screening Assays and 4.2 for Tools Compounds, making 4.1.1.-4.1.3. sub-subsections for the different assays (i.e., biochemical, computational and cell-based ones). Additionally, in the current sub-subsection 4.4.1., the last part of the paragraph mentions the use of ATG4B inhibitors, where this fragment is about ATG4B activators, are these statements correct or is wrongly mentioned? Moreover, figure 3 quite chaotic, only mentioning which are biochemical and cell-based techniques in the figure legend, but also figures 3 and 4 are partially redundant. It would be pertinent to rearrange figure 4 to clarify why is important the interconnections between the different boxes.
5.- As known, autophagy is important as a homeostatic mechanism. Authors must clarify their point of view on how ATG4B can be a druggable cancer target, being selectively targeted to the cancer cells, and not ubiquitously distributed into the whole body of human patients for the effectiveness of the treatment and minimization of potential side effects.
6.- Although authors mostly explain all the acronyms, some of them are missed and might be explained as well (i.e., ATG, MAP1LC3, GABARAP, PDAC, CML, FDA, LOPAC, GFP, CFP, YFP...).
7.- Some typographical mistakes and wrong nomenclature were found in the manuscript (in vivo, in vitro, ex vivo, neuro-degenerative, vps34, LC-II…).
Author Response
We thank the reviewer for these useful comments. Please see our response below.
1.- In the “Introduction” section, authors explain the different steps of autophagy pathway, and where LC3 appears and contributes to it. In Figure 1, authors should mention the distinct stages of this degradative pathway as well.
Response: We have updated Figure 1.
2.- In Figure 2, authors dissect how LC3 is processed by ATG4B. They should correct the nomenclature, because LC3B isoform is not the only isoform conjugated/deconjugated via this system, but the whole LC3/GABARAP family, as nicely described in the text of the manuscript. Moreover, in the same figure, it is mentioned LC3B-PE and LC3B-II below as there are 2 different conjugated-LC3B forms, providing some confusions to the reader.
Response: We have updated Figure 2.
3.- In the part 3 of the review, authors deeply describe the role of ATG4B in several forms of cancer. Since ATG4B has a different role depending where the cancer is located, it would be nice that they prepare an extra figure showing the distribution of the explained cancer and how ATG4B is relevant in each of them.
Response: This is a very interesting suggestion and hypothesis. However, there is not enough evidence in the literature to enable us to prepare such a figure. Most studies are based on xenograft models, which have very limited value for assessing the contribution of ATG4B with regards to cancer location. We have, however, included Table 2 on Biomarkers and In vivo models in the new version of the manuscript, which summarises all we could find related to the question of the in vivo relevance of ATG4B inhibition.
4.- Authors should improve the section 4 of the review (Screening Assays and Tool Compounds). It is suggested that they define a subsection 4.1. for Screening Assays and 4.2 for Tools Compounds, making 4.1.1.-4.1.3. sub-subsections for the different assays (i.e., biochemical, computational and cell-based ones). Additionally, in the current sub-subsection 4.4.1., the last part of the paragraph mentions the use of ATG4B inhibitors, where this fragment is about ATG4B activators, are these statements correct or is wrongly mentioned? Moreover, figure 3 quite chaotic, only mentioning which are biochemical and cell-based techniques in the figure legend, but also figures 3 and 4 are partially redundant. It would be pertinent to rearrange figure 4 to clarify why is important the interconnections between the different boxes.
Response: We have updated the sections as suggested. We agree that Figures 3 and 4 might have been confusing. We have updated Figure 3 with better headers. We have removed Figure 4 and instead included Tables 1 and 2, which summarise the key points of Figure 4 in a more concise way. The last section in 4.4.1. (now 4.2.2.) has been corrected to state “activators and inhibitors”. This paragraph is a conclusion of the section 4.2. and has been labelled as such.
5.- As known, autophagy is important as a homeostatic mechanism. Authors must clarify their point of view on how ATG4B can be a druggable cancer target, being selectively targeted to the cancer cells, and not ubiquitously distributed into the whole body of human patients for the effectiveness of the treatment and minimization of potential side effects.
Response: This is a very good point. We have described in the manuscript potential predictive and prognostic biomarkers and included this in Table 2, which might address how ATG4B and autophagy inhibition specifically reduces tumour growth in vivo. The point as to whether ATG4B inhibitors need to be targeted to the cancer is another question and we do not think that we are qualified to comment on this. At this point we could only speculate on how to achieve this.
6.- Although authors mostly explain all the acronyms, some of them are missed and might be explained as well (i.e., ATG, MAP1LC3, GABARAP, PDAC, CML, FDA, LOPAC, GFP, CFP, YFP...).
Response: Thank you very much for spotting this! We have updated the abbreviations in the text.
7.- Some typographical mistakes and wrong nomenclature were found in the manuscript (in vivo, in vitro, ex vivo, neuro-degenerative, vps34, LC-II…).
Response: Thank you very much! We have updated the typos and nomenclature.